# Chromatin maturation of the HIV-1 provirus in primary resting CD4$^{+}$ T cells

**Birgitta Lindqvist[1], Sara Svensson Akusjärvi[2], Anders Sönnerborg[2,3], Marios Dimitriou[4], J. Peter Svensson[1]***

**1** Department of Biosciences and Nutrition, Karolinska Institutet, Huddinge, Sweden, **2** Division of Clinical Microbiology, Department of Laboratory Medicine, Karolinska Institutet, Huddinge, Sweden, **3** Department of Medicine Huddinge, Division of Infectious Diseases, Karolinska Institutet, Huddinge, Sweden, **4** Department of Medicine Huddinge, Center for Hematology and Regenerative Medicine, Karolinska Institutet, Huddinge, Sweden

* peter.svensson@ki.se











**Data Availability Statement:** All sequencing files (H3K27ac ChIP-Seq) are available from the GEO database (accession number GSE121055, https://

## Abstract

Human immunodeficiency virus type 1 (HIV-1) infection is a chronic condition, where viral DNA integrates into the genome. Latently infected cells form a persistent, heterogeneous reservoir that at any time can reactivate the integrated HIV-1. Here we confirmed that latently infected cells from HIV-1 positive study participants exhibited active HIV-1 transcription but without production of mature spliced mRNAs. To elucidate the mechanisms behind this we employed primary HIV-1 latency models to study latency establishment and maintenance. We characterized proviral transcription and chromatin development in cultures of resting primary CD4$^{+}$ T-cells for four months after *ex vivo* HIV-1 infection. As heterochromatin (marked with H3K9me3 or H3K27me3) gradually stabilized, the provirus became less accessible with reduced activation potential. In a subset of infected cells, active marks (e.g. H3K27ac) and elongating RNAPII remained detectable at the latent provirus, despite prolonged proviral silencing. In many aspects, latent HIV-1 resembled an active enhancer in a subset of resting cells. The enhancer chromatin actively promoted latency and the enhancer-specific CBP/P300-inhibitor GNE049 was identified as a new latency reversal agent. The division of the latent reservoir according to distinct chromatin compositions with different reactivation potential enforces the notion that even though a relatively large set of cells contains the HIV-1 provirus, only a discrete subset is readily able to reactivate the provirus and spread the infection.

## Author summary

HIV infection is a devastating disease affecting 35 million people worldwide. Current anti-retroviral treatment is highly effective and has made the HIV infection chronic. However, despite more effective treatments, the prospects of a cure are distant. The problem for an HIV cure is that, even though the virus particles are eradicated, the infected cells maintain the information of remake the virus. This information is integrated in the host cell as a provirus. The provirus switches between active and inactive states. Thereby, the infected cells evade both the immune system and death associated with massive viral

www.ncbi.nlm.nih.gov/geo/query/acc.cgi?acc=GSE121055).

**Funding:** Vetenskapsrådet-M (2015-02312, 2019-00991, https://vr.se), Cancerfonden (2016/576, 19 0412 Pj, https://www.cancerfonden.se), Åke Wibergs Stiftelse (M14-0205, M15-0044, M16-0022, http://ake-wiberg.se), Clas Groschinskys Minnesfond (M1662, M1790, http://www.groschinsky.org), and Läkare mot AIDS Forskningsfond (FOb2017-0001 http://www.aidsfond.se), awarded to J.P.S.; and Vetenskapsrådet-M (2016-01675, https://vr.se) and Karolinska Institutet (KID-grant 2-1930/2016, http://ki.se) awarded to A.S.; and from CIMED and Region Stockholm (Project 20190092, http://cimed.ki.se), awarded to A.S. and J.P.S. The funders had no role in study design, data collection and analysis, decision to publish, or preparation of the manuscript.

**Competing interests:** The authors have declared that no competing interests exist.

production. We have characterized the composition of proviral chromatin and how it connects with transcription and viral production. In resting primary CD4+ T-cells, we follow the fate of the provirus starting at infection until latency is firmly established. Only in a fraction of intact proviruses were we able to reverse latency and that this was highly regulated by the chromatin composition. Whereas the proviruses encompassed in heterochromatin were refractory to activation, latent proviruses with "enhancer" characteristics were readily activated. Our study provides key insights as to detect the remaining HIV-1 infected cells capable of reseeding the infection, and the mechanisms whereby they are maintained.

## Introduction

Once human immunodeficiency virus type 1 (HIV-1) infects a cell, typically an activated $CD4^+$ T-lymphocyte, the viral genome can integrate into the host chromosome as a provirus. Upon integration, the viral sequence is packaged into chromatin, and in a subset of cells, as the cell returns to quiescence, the proviral chromatin becomes condensed and silenced together with large portions of host chromatin [1,2]. At any time can the provirus reactivate, leading to viral gene expression and virus production. The reservoir of rare cells harboring a latent intact provirus, 1 in $10^5$–$10^6$ circulating $CD4^+$ T-cells [3,4], is considered the main obstacle for curing HIV-1/AIDS. Latently infected cells are, to the best of our current knowledge, indistinguishable from uninfected cells. Consequently, the virus escapes the immune system and the actions of current anti-retroviral drugs. A means for identifying this functional reservoir would represent a milestone in clinical practice.

The latent reservoir displays heterogeneity in many aspects, such as the anatomical location in the body, the types of cells infected, and the strength of silencing. Also, the reservoir harbors only a small fraction (5–12%) of functional, intact proviruses [5–10]. The majority of proviruses contain big internal deletions, or they have pre-mature stop codons induced by apolipoprotein B mRNA editing enzyme (APOBEC) catalytic subunit 3G [11]. Both of these mutational events typically occur before or during the integration of viral DNA. Also other events can lead to non-functional proviruses. Failure to produce viruses can be a result of e.g. variations in functions of viral proteins or host mechanisms leading to ineffective RNA processing, co-factor sequestering or epigenetic silencing [12–19]. Even though defective proviruses cannot produce infective particles, they can be transcribed resulting in both mature spliced RNAs and proteins, acting as decoys to the immune system, and thus, they reshape the reservoir [20]. This leads to a continuous evolution of the viral reservoir in a person living with HIV (PLWH) [21].

The activation potential of the provirus depends on several factors, including the epigenetic context and the nuclear environment [22–26]. Initially, HIV-1 is guided to active regions by the viral integrase that interacts with the host factor lens epithelium-derived growth factor (LEDGF). LEDGF recognizes histone H3 trimethylated at lysine 3 (H3K36me3) found in transcribed regions and H3K4me1 found at enhancers [27]. Among productively infected cells and reactivated latently-infected cells in the reservoir, proviruses are mainly found in generally active or poised chromatin at gene coding regions or enhancers. In contrast, permanently silenced proviruses are predominantly found in regions of heterochromatin [22].

As infected cells return to quiescence, the inactive proviral chromatin frequently acquires repressive histone modifications. Polycomb repressive complex 2 (PRC2) and euchromatic histone-lysine N-methyltransferase 2 (EHMT2) are recruited to the proviral chromatin [28,29]

leading to accumulation of facultative heterochromatin mark H3K27me3. ABOBEC subunit 3A recruits KRAB-associated protein 1 (KAP1) and heterochromatin protein 1 (HP1) [30] for the constitutive heterochromatin mark H3K9me3 [31–35]. In this manner, HIV-1 proviral silencing is established and maintained. A lack of H3K27me3, H3K9me2/3, or DNA methylation sensitizes latent proviruses to latency reversal agents (LRAs) [28,29,36,37]. The process of establishing heterochromatin is lengthy and complex [38]. Upon HIV-1 infection, different epigenetic marks are initially established over the provirus, but the H3K27me3-to-H3K9me3 ratio evolves over time [39]. Due to the compact nature of heterochromatin structures, access to the transcriptional machinery at the canonical long-terminal repeat (LTR) promoter becomes restricted, and thus, HIV-1 transcription is believed to be hampered and HIV-1 latency is promoted [18,35]. However, RNA polymerase II (RNAPII) and active chromatin marks, such as H3K4me3, have been shown to remain on the LTR promoter, maintaining the promoter in a state poised for transcription. The transcription start site (TSS) for all viral transcripts resides within the LTR.

The post-initiation block of RNAPII has long been recognized as a rate-limiting step of HIV-1 latency reversal [40]. This block is alleviated by binding of the viral transactivating regulatory protein Tat to the promoter-proximal trans-activation response element (TAR) [41]. However, recent data have also highlighted the roles of blocks in elongation, splicing, and termination [13]. Cellular HIV-1 transcription-restrictive factors are important to prevent viral replication. Incomplete HIV-1 transcripts of varying lengths have been found in latently infected cells [13,42–44]. From a clinical perspective, these transcripts appear to be important as they negatively correlate with time to HIV-1 rebound after ART interruption [45].

A strong stimulus for reversing proviral latency is T-cell stimulation. However, only few infected cells (<5%) display proviral activation upon a single round of *ex vivo* T-cell stimulation [22]. Both T-cell stimulation and LRA administration have been shown to elevate viral RNA levels, but these treatments have modest effects on reduction of the latent reservoir in PLWH [37,46–49].

Here, we dissected the chromatin and RNA landscape of the HIV-1 provirus during latency establishment in primary resting $CD4^+$ T-cells. We aimed to reveal mechanisms that maintain the activation potential of latently infected cells.

## Results

### Transcription over the provirus isolated from cells from PLWH under ART

To confirm that the HIV-1 provirus was transcribed in cells from PLWH successfully treated with ART [13], we isolated RNA from peripheral blood mononuclear cells (PBMCs) from ten PLWH with HIV-1 of diverse subtypes. These ten study participants were all successfully treated with plasma virus levels <50 copies/μl for a median of 8 years (range 1.8–22.3 years) and $CD4^+$ T-cells within the normal range (S1 Table). The levels of cell-associated (CA) RNA were measured after reverse transcription (RT), followed by droplet digital PCR (ddPCR). Several well-documented primer pairs were tested over the proviral genome (S1A) to determine the level of transcription at different proviral sites: read-through transcription at the 5´ LTR upstream of the TSS, *TAR*, *nuc-1* (also known as *long LTR*), *gag*, *pol*, *env*, *nef*, as well as polyadenylated (*polyA*) and multiply spliced (*tat-rev*) transcripts. Unspecific background signal was quantitated in RNA from $CD4^+$ cells from four HIV-negative donors, revealing a specific signal for all except perhaps the *polyA* probe (S1B Fig). The primers were tested for efficiency in each patient and cell type by amplifying the primer pairs against the genomic DNA (S1C Fig). Points were discarded where DNA could not be amplified. The *tat-rev* and *polyA* primers could not be tested here as they target the mRNA with no DNA template. In unstimulated

PBMCs, viral CA-RNA levels at *TAR* and *nuc-1* were significantly ($p<0.01$) higher than background (*read-through*) levels of transcription (Fig 1A). In turn, the abundance of mature multiply spliced transcripts (*tat-rev*) was significantly lower ($p<0.05$) than the abundances of both *TAR* and *nuc-1* products. We isolated resting $CD4^+$ cells, the specific cell type thought to harbor the latent reservoir, from a subset of the same blood samples (Fig 1B). Also in this cell population, the multiply spliced transcripts were detected at lower levels ($p<0.05$) than initiated transcripts from the *TAR* region. The remaining $CD4^+$-depleted cells were analyzed, where we could detect proviral DNA, but the RNA signals fell within the background (S1D Fig). In most study participants, the highest RNA levels originated from the *TAR* and *nuc-1* region of HIV-1. This indicated that transcription in latent proviruses had initiated RNAPII elongation to a low degree, but still, mature transcripts failed to emerge.

## *In vitro* infection of primary $CD4^+$ T cells

To enable more detailed analysis, we turned to models for HIV-1 latency in primary human $CD4^+$ T cells, as technical and ethical issues hindered molecular characterization in cells from PLWH. We isolated $CD4^+$ cells from fresh peripheral blood collected from four HIV-negative donors. We adapted a protocol to study quiescent effector memory T ($T_{EM}$) cells enriched for latent HIV-1 [1]. Upon isolation, the $CD4^+$ cells were immediately stimulated with antibodies against CD3 and CD28 (αCD3-CD28) and grown in media containing growth factors (Fig 1C). On day three, the activated cells were transferred to cytokine-free media as to return to quiescence and transition into effector memory ($T_{EM}$) cells. On day six, the transitioning cells were infected with a HIV-1 reporter virus. The HIV-1 reporter virus encoded the full-length viral genome that had been rendered replication-deficient with six mutations. In addition, the *GFP* gene was inserted in the *env* coding region. On day ten, cells were harvested. The viability range was 46–78% as reported by flow cytometry. HIV-1 integration was observed in 12±3% of the cells as assessed by DNA isolation followed by ddPCR using primers of the 5′ LTR [50,51]. The unspecific background signal here was estimated in cells from the same donor before *in vitro* infection to 0.5±0.4%. As control, we included the J-lat clone 5A8, a Jurkat-derived cell line with one integrated latent HIV-1 reporter provirus per cell [24,52], where integration was estimated to 96±6%.

At day ten, we also isolated RNA from the HIV-1 infected $T_{EM}$ cells. By RT-ddPCR, we quantitated transcript levels at four proviral regions: *read-through*, *gag*, *pol* and *tat-rev* (Fig 1D). Transcription from the *gag* region was significantly ($p<0.05$) higher than transcription from the other three regions. This observation suggested that transcription had been initiated and some elongation was in progress, but with abortive long-range elongation and no production of mature transcripts. Similarly, in latent 5A8 cells 5′ transcripts were detected but no processed spliced transcripts (Fig 1E). Upon T-cell stimulation with phorbol 12-myristate 13-acetate, in conjunction with ionomycin (PMA/i), transcription increased dramatically. The strongest relative increase was observed in the multiply spliced transcripts (*tat-rev*).

A lack in RNA processivity and transcript maturation in latently infected primary $T_{EM}$ cells is further substantiated by the fact that, despite >10% of the $T_{EM}$ cells having integrated HIV-GFP, the GFP signal could only be detected in 0.1±0.04% of unperturbed or DMSO treated cells at day ten (Fig 1F). We stimulated the cells with PMA/i and also exposed them to a panel of LRAs, consisting of the HDAC inhibitors panabinostat and romidepsin, the protein kinase C agonist bryostatin, the BET inhibitor JQ1 and inhibitor of DNA methylation 5-aza-deoxycytidine. These drugs were administered in single regimens or in combinations, and in the presence of raltegravir to hinder viral reintegration. The largest effect of the LRAs in this primary T-cell model, as measured by percentage of GFP-positive cells, came from the HDAC

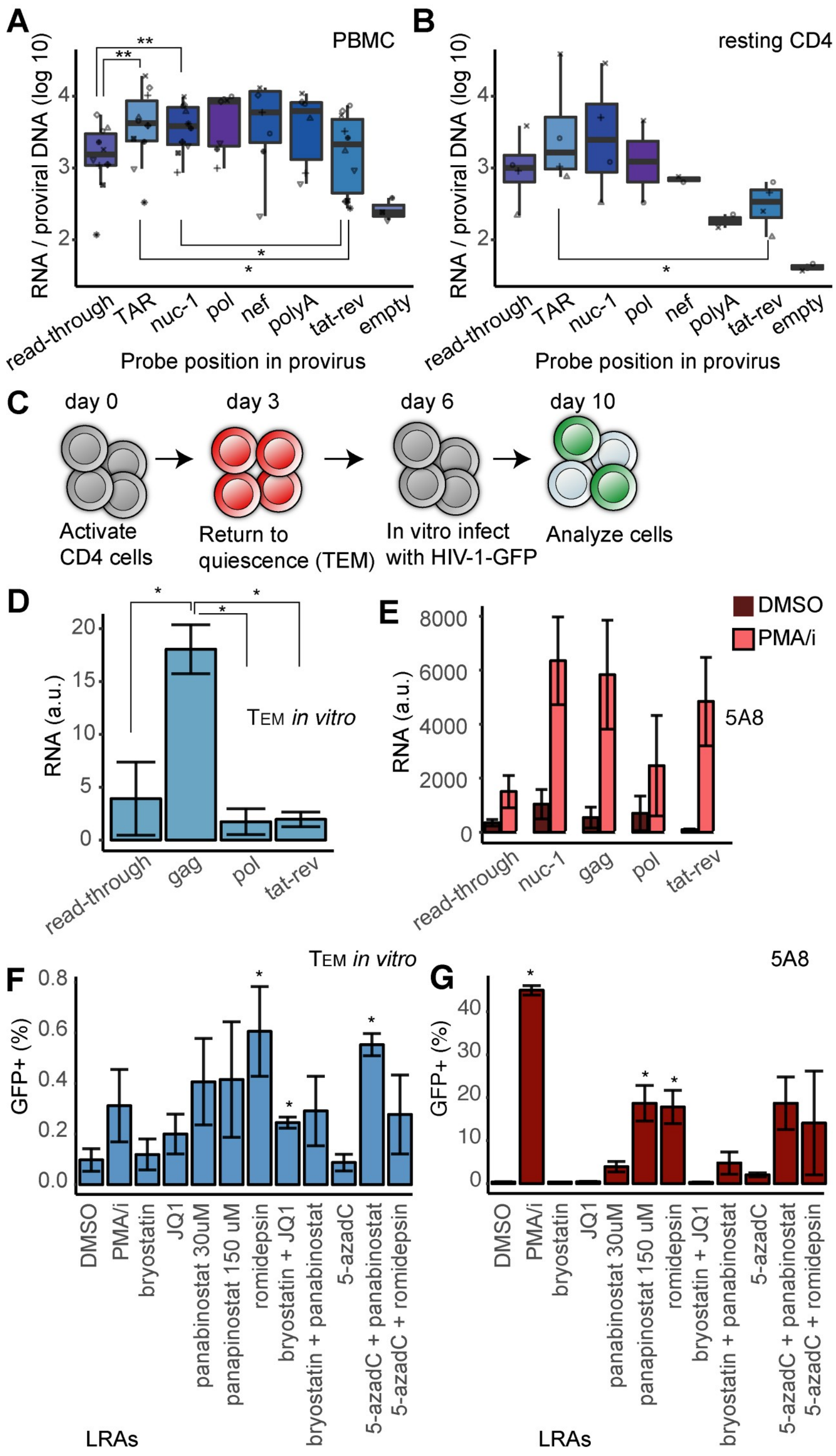


**Fig 1. Transcripts in cells from HIV positive study participants and infection of primary $T_{EM}$ cells.** (**A**) Box plot showing transcription levels in PBMCs from well-treated PLWH (*n* = 10), at various positions of the provirus. Each study participant is represented by a unique symbol. The median signal is represented by the thick line and the box covers the 25 to 75 percentile of the values. P-values calculated by the Wilcoxon signed rank test. (**B**) Transcription levels in resting CD4+ T-cells from a subset of PBMCs in panel A (*n* = 5), at various positions of the provirus. Each study participant is represented by a unique symbol. P-values calculated by the Wilcoxon rank sum test. (**C**) Schematic of the generation of HIV-1 $T_{EM}$ cells. (**D**) Transcription levels in primary $T_{EM}$ cells HIV-1 infected *in vitro* (*n* = 4). (**E**) Transcription levels of HIV-1 in J-lat 5A8 cells after T-cell activation (PMA/i) or mock-treated (DMSO) (*n* = 5). (**F**) HIV-1 latency reversal in $T_{EM}$ cells after 24h exposure to a panel of putative latency reversal agents. (**G**) HIV-1 latency reversal in J-lat 5A8 cells after 24h exposure to the panel of molecules. For panels (D), (F), (G) p-values were calculated by unpaired t-test, $^{*}p<0.05$, $^{**}p<0.01$.


inhibitors panabinostat and romidepsin. Pretreatment with 5-aza-deoxycytidine 3 hours before HDAC inhibition showed no significant effect on latency reversal compared to HDAC inhibition alone. A similar latency reversal profile was identified in the J-lat 5A8 cells (Fig 1G). Notably, the amplitude of latency reversal, scored as the GFP intensity within the GFP-positive cells was considerably higher in PMA/i treatment than after the other treatments (S2 Fig).

Together, these data reveal that viral transcription extends beyond the promoter in latent cells, but it fails to produce mature transcripts.

## A primary cell model to study proviral latency maturation

To increase the *in vitro* life span of primary $CD4^+$ cells and enable the study of proviral latency maturation, we made use of the Bcl2-model system [53,54]. In the Bcl2-model, $CD4^+$ lymphocytes are made to express the anti-apoptotic protein Bcl2. Ectopic Bcl2 expression result in cells that can be cultured for extended periods of time (Fig 2A). Here, $CD4^+$ cells were isolated from fresh peripheral blood collected from three HIV-negative donors. Upon 72 hours αCD3-CD28 stimulation, the $CD4^+$ cells were transduced with a lentiviral vector that carried the *bcl2* gene. Cells were then transfer to cytokine-free media as to return to quiescence. After three weeks, the cells were either cryopreserved or re-stimulated with αCD3-CD28 for 72 hours. Activated cells were infected with a HIV-1 reporter virus and maintained under stimulating growth conditions for an additional three days. Bcl2-model cells isolated from the three HIV-negative individuals were divided into two groups; one was infected with HIV-1 immediately, and the other was infected after one freeze-thaw cycle.

To determine the fraction of cells initially *in vitro* infected by HIV-1, we isolated DNA at three days post infection (dpi). The lentiviruses had been nearly exclusively integrated, as 2-LTR circles were not detectable by Taqman PCR. We quantified the levels of integrated proviral DNA with ddPCR and three probes that targeted *gag* and *env* (Fig 2B), as well as the 5′ LTR [50,51]. For comparison, we included the J-lat clone 5A8. Among the Bcl2-model cells, the percentage of successful HIV-1 infections ranged from 3.1% to 18.2% (median: 10.8%), estimated with the standard *gag* probe. Similar results were obtained using the *env* probe (median: 9.8%) (Fig 2C).

The 5′ LTR probe signal was detectable at higher levels (median: 19.5%) compared to *gag* and *env* probes. As the primary cells were infected with two lentiviruses, encoding the Bcl2 construct and the HIV-1 construct, both containing a 5′ LTR sequence, this increased signal was expected compared to probes unique to HIV-1. However, also large internal deletions in the provirus result in a lower signal for the *env* and *gag* primers relative to *5′ LTR*.

## Intact provirus in the majority of cells

Given the large potential of mutations occurring prior to integration, we sought to determine the initial fraction of intact proviruses in our model cells. Large deletions were estimated by the imbalance between the 5′ *gag* and internal *env* signals. The infected cells showed no significant bias towards any of the deletions (Fig 2D). To calculate the fraction of hypermutated proviruses, we relied on the lower signal produced by ddPCR after APOBEC3G-induced G-to-A mutations. These resulting point mutations produce a mismatch with the primers or probe that lead to reduced efficiency of the PCR reaction. The specific *env* probe used here had been designed to target an APOBEC3G hotspot as the primers and probe matched a cluster of 12 previously described APOBEC3G-induced mutations [55]. Amplification of mutated proviral sequences resulted in the characteristic "rain" pattern within the cluster of $env^+$ droplets (Fig 2B, most pronounced in donor II). The donor cells showed substantial levels of hypermutations, ranging from 26% to 33% (Fig 2E). The 5A8 cells were used to confirm that virtually no

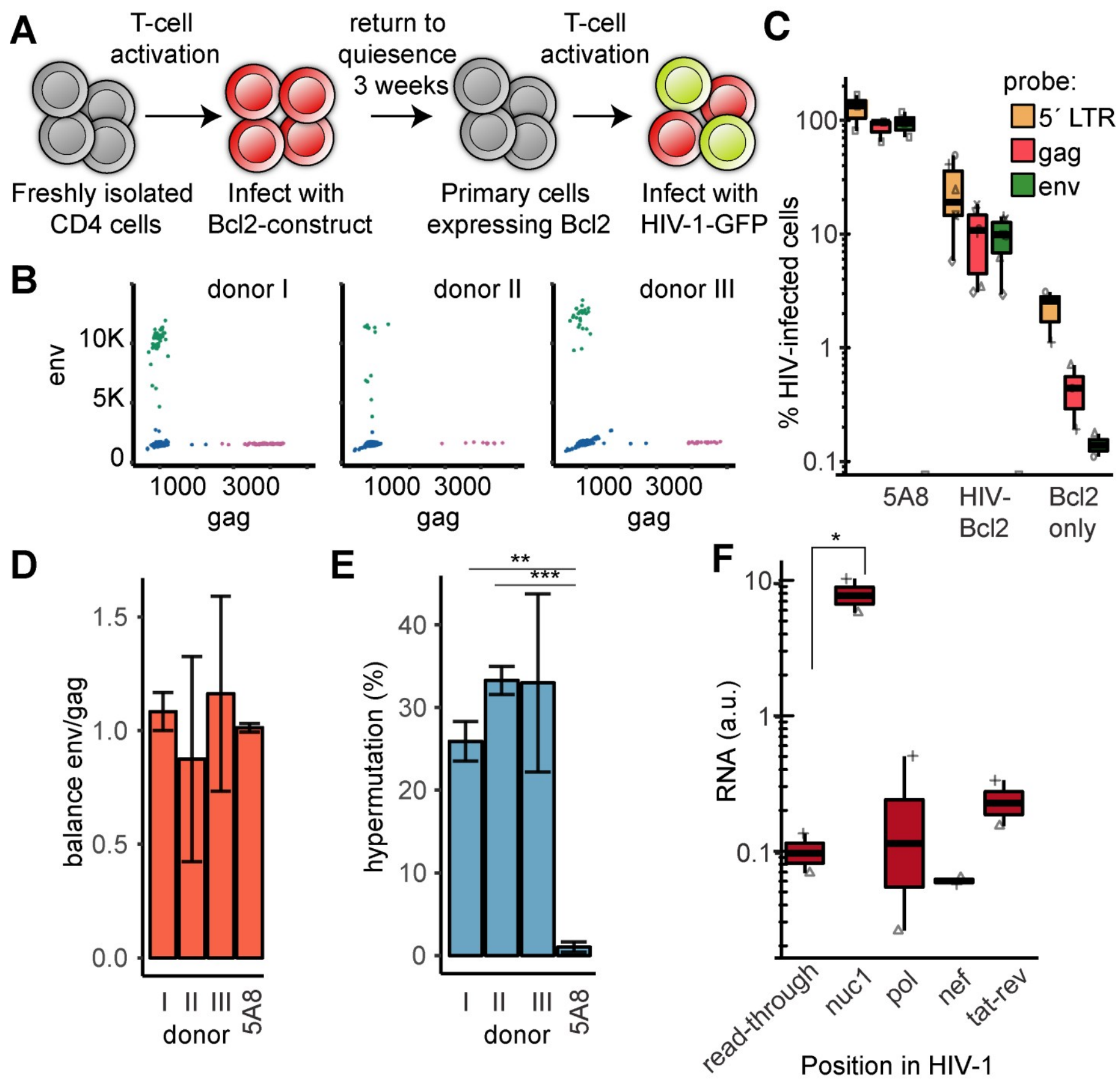


**Fig 2. Primary cells to study long-term HIV-1 latency establishment.** (**A**) Schematic of the generation of HIV-1 carrying Bcl2 model cells. (**B**). ddPCR scatterplots of HIV-1 infected Bcl2-cells from three HIV-negative donors with probes against *env* (y-axis) and *gag* (x-axis). DNA isolated at 3 dpi. (**C**) Quantification of ddPCR results over three probes (*5´LTR*, *gag* and *env*) in control cell line J-lat 5A8, and primary cells from the three donors in biological duplicate, either with both Bcl2 and HIV-1 or Bcl2-only. The data from individual donors and duplicates are visualized by differently shaped points. (**D**) Ratios of *gag*/*env* ddPCR signals to estimate internal 5´and 3´ HIV-1 deletions in the three donors and the control cell line J-lat 5A8. (**E**) Fraction of "rain", i.e. low *env* signals (approximately below 10,000 a.u.) reflecting APOBEC3G-induced hypermutations. (**F**) RT-ddPCR results of cell-associated RNA isolated at 3 dpi ($n = 2$). Probes were as in previous results. *p<0.05, **p<0.01, ***p<0.005 calculated by unpaired t-test in panels (E) and (F).


rain (0.2%) could be detected in settings without hypermutations. In summary, even though a majority of the proviruses in the model cells are intact, a significant fraction have APOBEC3G-induced point mutations. We also measured RNA levels in unperturbed model cells. CA-RNA was isolated and following RT-ddPCR, we found a significantly ($p<0.05$) elevated *nuc1* RNA signal compared to read-through (Fig 2F).

## Number of HIV-1 infected cells in time

At 72 hours after HIV-1 infection, we returned the cells to the resting state by transferring them to cytokine-free media (Fig 3A). The cells were maintained in culture and followed for four months. Samples were collected at 30, 50, 70, 90, and 120 dpi. The cultures remained stable although the viability decreased over time (S3A Fig). To determine changes in the number of HIV-1 infected cells, we quantified the fraction of cells that harbored the provirus at each time point by performing ddPCR on genomic DNA. We used the *gag* and *env* probes to

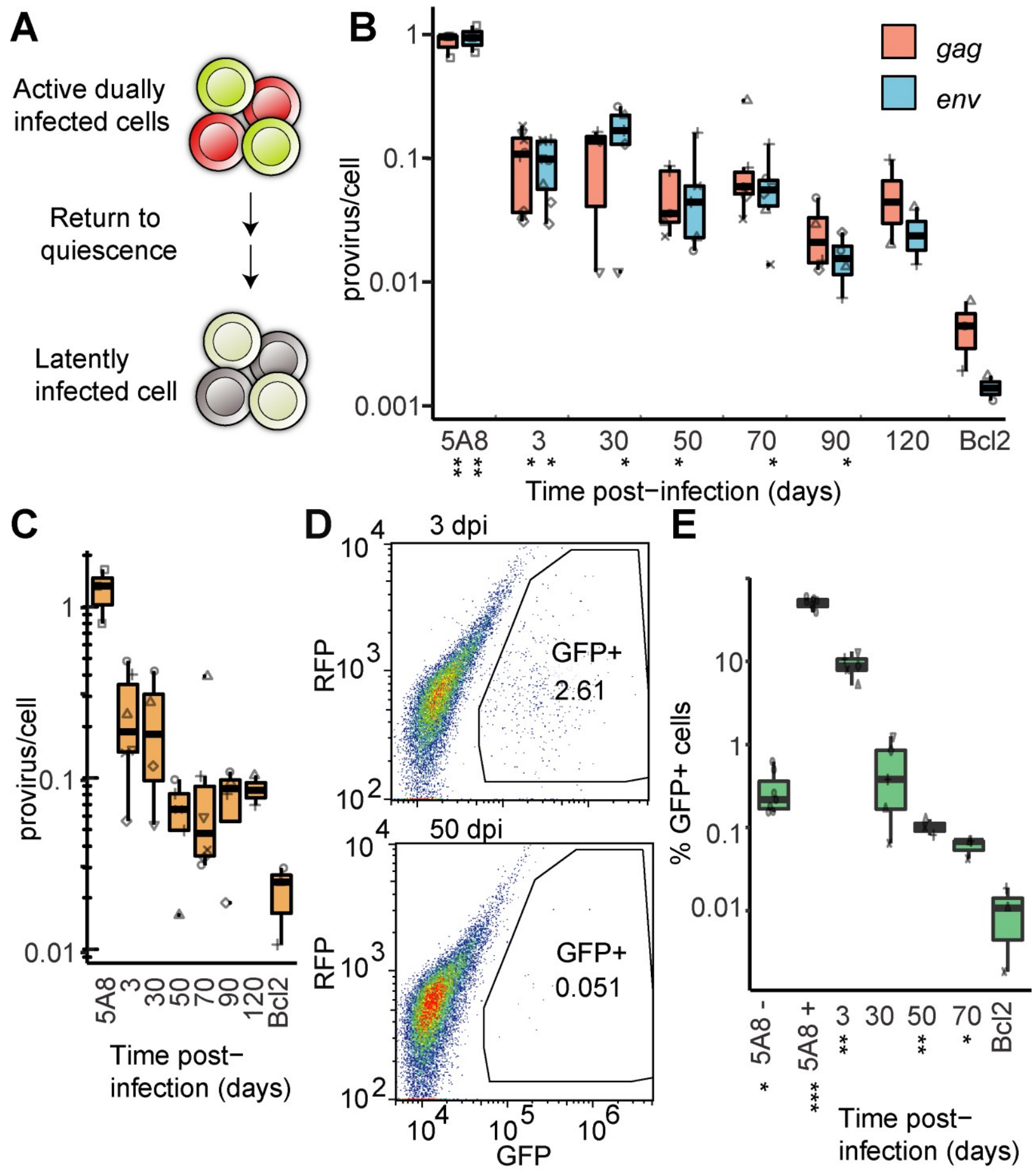


**Fig 3. Diminishing activation potential of the HIV-1 provirus in resting T-cells in time.** (**A**) Active cells were transferred to cytokine-free media to return to quiescence. (**B**) Proviruses quantified by ddPCR at the *gag* and *env* loci. Signals normalized to endogenous *rrp30*. J-lat clone 5A8 has a single integrated provirus and the Bcl2-only cells have been infected with the Bcl2-virus but not the HIV-reporter virus. (**C**) Proviruses quantified by ddPCR at the *5′ LTR* locus. Signals normalized to endogenous *rrp30*. (**D**) Flow analysis of cells at 3 and 50 dpi gated for GFP-positive cells expressing the GFP-containing HIV-reporter provirus. (**E**) Quantification of GFP positive cells in time, with addition of 5A8 treated with DMSO (5A8-) or PMA/i (5A8+) for 24 hours. *$p<0.05$, **$p<0.01$ (below the x-axis), calculated by unpaired t-test in panels (B) and (E), comparing each probe and time-point to the Bcl2 control.


identify HIV-1 unique regions (Fig 3B), and 5′ LTR probes to detect both the HIV-1 and the integrated Bcl2 segment (Fig 3C). At 90 dpi, the estimated fraction of HIV-1 infected cells was 2.2% (median *gag*) or 1.6% (median *env*). The 5′ LTR signal leveled out over time; at 90 dpi, the signal indicated 9.0% (median) remaining infected cells, which suggested that maximally 7% of cells contain the *bcl2* gene. The data show that throughout our experiment, the HIV-1 provirus remains present and that the cultures slowly evolve after the initial latency establishment.

## Spontaneous HIV-1 activity decreases in time

Next, we determined the prevalence of productively infected cells over time. GFP-positive cells in unperturbed cultures were counted with flow cytometry (Fig 3D). Initially at 3 dpi, 5.2–13.1% of cells were $GFP^{+}$. However, we could not detect significant RNA levels downstream of

nuc-1 (Fig 2F), possibly due to technical issues. In time, the GFP frequency declined substantially but remained detectable and distinguished from the parental Bcl2-only negative control even at 70 dpi ($p<0.05$) (Fig 3E). This decline is expected as more infected cells transitioning from productive to latent infection. A decline would also be expected from provirus acquiring a "deeper" latency [26], eliminating the possibility of spontaneous reactivation.

As an alternative method for tracking provirus activity, we measured RNA levels. However at 50 dpi, transcription could not be reliably detected above the read-through level, possibly for technical reasons (S3B Fig). The quality of RNA in this experiment suffered from low viability of the cell culture.

## Prolonged quiescence diminished proviral reactivation potential

To examine the mechanisms underlying proviral activation, we activated T-cells with αCD3-CD28 or PMA/i (Fig 4A). Activation was performed in the presence of raltegravir to hinder viral reintegration. The T-cell activation was phenotypically assessed and confirmed by accelerated cell growth, more pronounced cell lumpiness, rapid media turnover, and increased cell death. Unexpectedly, these clearly activated cells largely failed to present the predicted surface markers (CD25 or CD69) indicative of T-cell activation (S4 Fig). In contrast, the control J-lat 5A8 cells presented both CD25 and CD69 after activation.

To detect latent HIV-1 reversal, we again quantified CA-RNA with RT-ddPCR throughout the viral transcript. At 50 dpi, the mature multiply-spliced transcripts were 10±3 fold increased ($p<0.05$) after 48 h of T-cell stimulation (Fig 4B). The *nef* probe, which detected all transcripts completed to the 30078 end, showed a similar increase in transcription (9±3 fold increase, $p<0.05$). The *nuc1* and *pol* probes at the 5´ region showed no increase after T-cell activation, consistent with an ongoing, non-productive transcription at the 5´ region marginally affected by T-cell stimulation in primary cells [13].

To determine the number of cells with activated provirus as a consequence of T-cell stimulation, GFP-positive cells were monitored by flow cytometry at three time-points (30, 50, and 70 dpi; Fig 4C). Here we analyzed viable cells only, as addition of a membrane-permeable dye

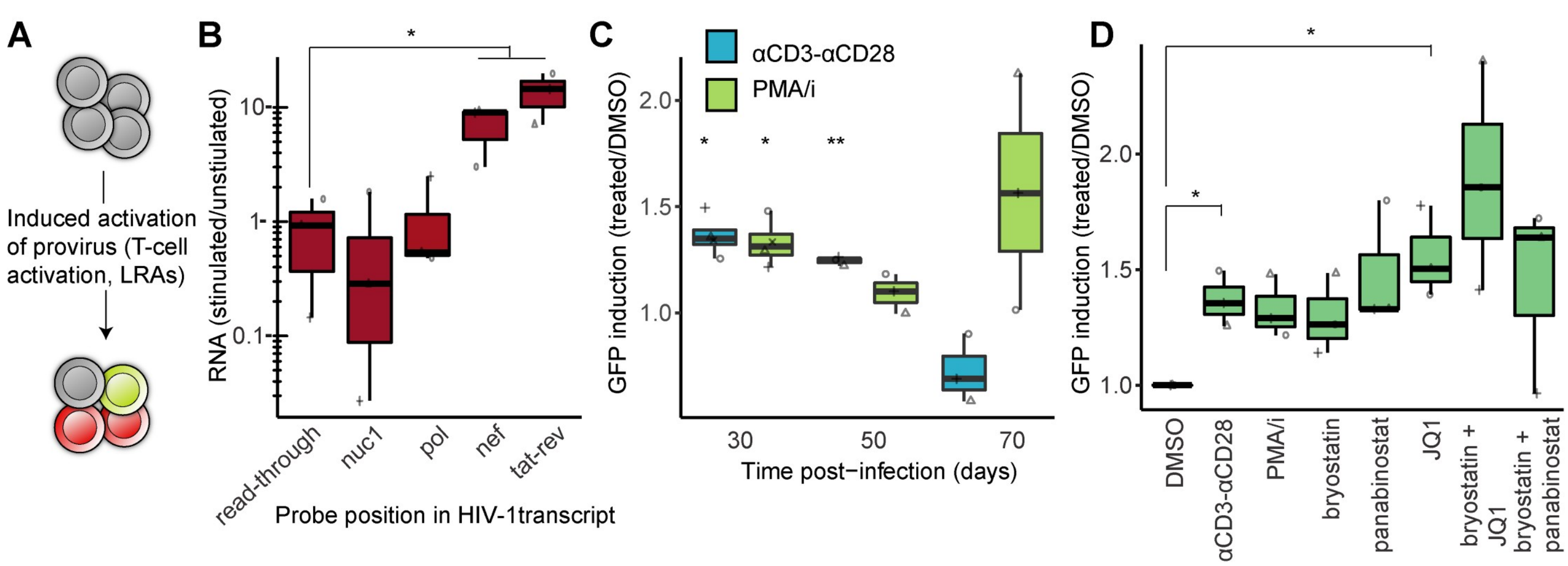


**Fig 4. HIV-1 latency reversal by T-cell activation and LRAs.** (**A**) Resting cells were treated with αCD3-CD28, PMA/i, or LRAs. (**B**) RT-ddPCR results at 4 proviral loci and the splice product *tat-rev* in activated/resting cells. (**C**) GFP-positive cells gated in flow cytometry at 30, 50 and 70 dpi after 48h treatment with DMSO, αCD3-CD28, or PMA/i. P-values are calculated between treated (αCD3-CD28 or PMA/i) and mock-treated (DMSO) at each time-point. (**D**) GFP-positive cells gated in flow cytometry at 50 dpi after 48h treatment with LRAs. *$p<0.05$, **$p<0.01$ (below the x-axis), calculated by paired t-test in panels (B–D).


allowed discrimination between live and dead cells. T-cell activation using αCD3-CD28 conjugation and PMA/i resulted in small, but significant increases ($p<0.05$) in viable cells with activated provirus compared to DMSO-treated cells at 30 and, for αCD3-CD28, also at 50 dpi. At 70 dpi, no effect could be determined, possibly as a consequence of few GFP-positive cells in the unperturbed state and poor cell viability.

Next, at 50 dpi, we exposed cells for 48 hours to a subset of the previously described LRAs. Cell viability was determined and only after PMA/i could we detect a significant ($p<0.05$) negative impact on viability (S5 Fig). The basal level of GFP+ cells after DMSO treatment was very low (0.01–0.13% GFP+), and following PMA/i stimulation few cells reactivated HIV-1 (0.07–0.15% GFP+). The latency reversal results showed a large spread between individuals but only barely did any LRAs show a significant effect; only JQ1 alone or JQ1 together with bryostatin consistently induced some proviral activation (Fig 4D). This was in contrast to the results from the $T_{EM}$ model (Fig 1F) or the J-lat 5A8 cells (Fig 1G) where HDAC inhibitors showed the greatest potential for HIV-1 latency reversal.

## Inactive chromatin marks accumulate over the provirus

Upon integration, the proviral DNA sequence is rapidly encapsulated in chromatin. To follow the chromatin silencing of the provirus during latency establishment, we assessed the distribution of constitutive and facultative heterochromatin marks by chromatin immunoprecipitation (ChIP) using antibodies against H3K9me3 and H3K27me3, followed by quantitative PCR (qPCR). To prevent erroneous signal from dead cells, prior to chromatin isolation viable cells were isolated by Ficoll density centrifugation. Four additional primers spanning the HIV-1 provirus were used, querying again the first nucleosome after the TSS (HXB2 position 555–664), *gag* (pos 1299–1323), *pol* (pos 4058–4258) and *vpr* (5750–5772). The last two regions have previously been implicated in regulatory functions [56,57]. For simplicity we refer to the primer pairs as targeting nucleosome 1, 6, 25 and 35.

The constitutive H3K9me3-mark appeared to be distributed over the proviral body by 30 dpi (Fig 5A). However, the large variation between donors should be noted. In contrast, the facultative H3K27me3-mark gradually became more prominent throughout the time-course (Fig 5B). In J-lat 5A8 cells, a signal from H3K9me3 and H3K27me3 was detected but it was not significantly different from background ($p>0.05$).

Methylated H3K9 serves as a binding platform for heterochromatin protein 1 (HP1), found at the latent provirus [31]. HP1 family member α leads to chromatin compaction and inaccessibility, possibly by nuclear droplet formation through liquid-liquid phase separation [58]. To confirm proviral heterochromatin formation, we determined chromatin accessibility by treating isolated chromatin with a panel of nucleases and amplifying the resulting fragments with qPCR. As expected, in time the proviral chromatin became increasingly compact as its accessibility declined over time (Fig 5C).

## Elongating RNAPII at proviral chromatin

We then aimed to relate the proviral reactivation potential to the transcription machinery. RNAPII elongation has long been identified as a key limiting step in HIV-1 transcription [40]. Here, we measured two forms of RNAPII at the provirus during the establishment of latency. During RNAPII transcription initiation, the repetitive C-terminal domain of the RPT1 subunit becomes hyper-phosphorylated at serine 5 (ser5p). During transcription elongation also serine 2 of the same RPT1 domain becomes phosphorylated (ser2p) at an increasing degree. Over the HIV-1 provirus, we found that the signal for initiated RNAPII (ser5p) varied considerably, but was consistently found at the promoter region through out the experiment (Fig 6A). Also the

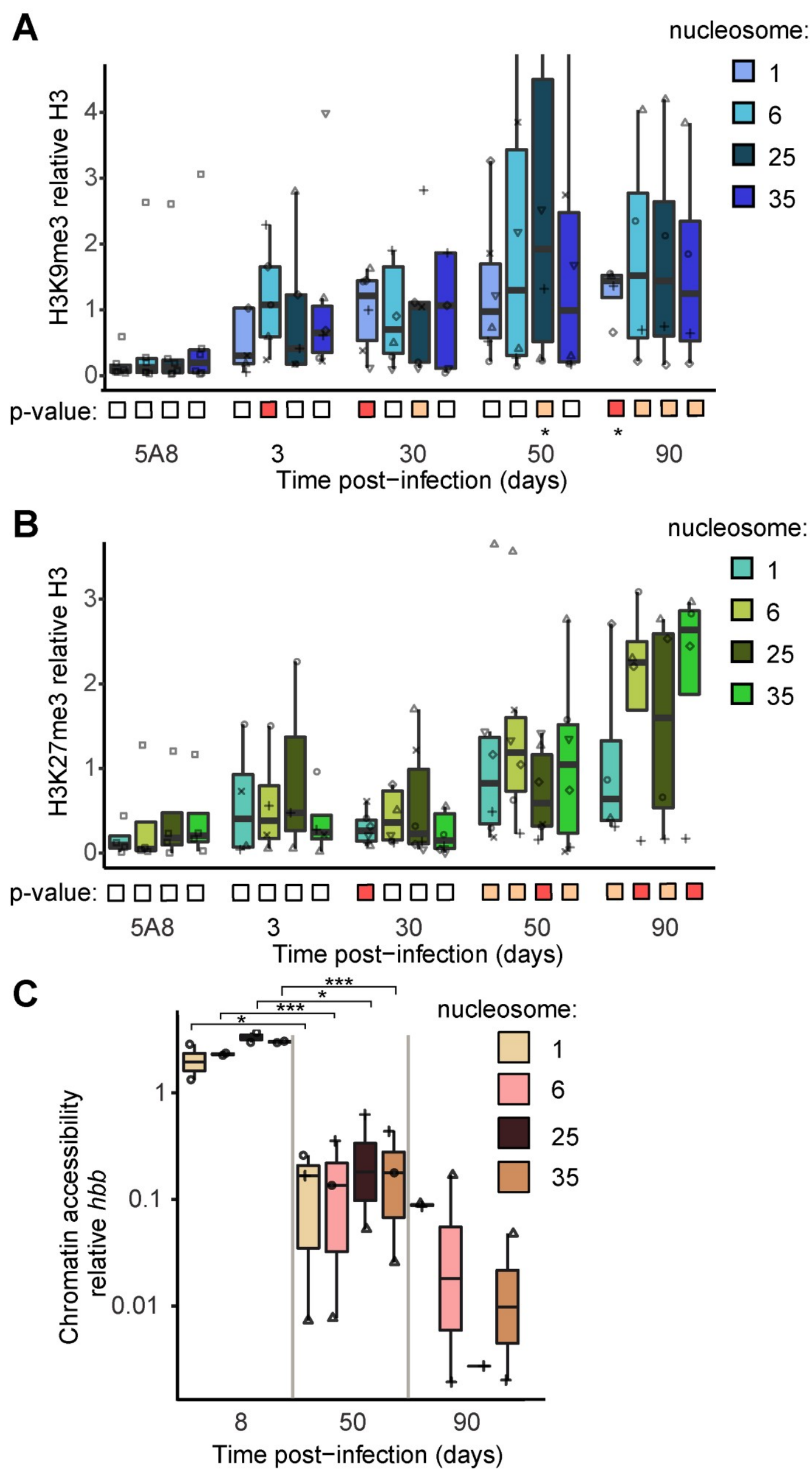


**Fig 5. Appearance of heterochromatin and chromatin compaction along HIV-1 proviral chromatin.** (**A**) Boxplots for ChIP-qPCR signals for constitutive heterochromatin mark H3K9me3. (**B**) ChIP-qPCR signal for facultative heterochromatin mark H3K27me3. In panels (A, B), for each point the specific ChIP signal is divided by the H3 signal for the same sample. Samples where the unspecific IgG signal is higher than the specific signal are discarded. P-values were calculated: boxes below the x-axis indicate a one-sample t-test (null hypothesis: specific normalized ChIP signal equals zero) orange: $p<0.05$, red: $p<0.01$, white: $p>0.05$, stars indicate significance using a two-sample t-test (null hypothesis: specific ChIP signal equals IgG signal) $^{*}p<0.05$, $^{**}p<0.01$. (**C**) qPCR signal revealing chromatin accessibility to nucleases. Data is normalized to the heterochromatin *hbb* locus of the same sample ($n = 3$). $^{*}p<0.05$, $^{***}p<0.001$ calculated by paired t-test.


elongating form of RNAPII (ser2p) was found, albeit at a lower degree, with the highest signal in proximity of the promoter (Fig 6B). As before, we used J-lat clone 5A8 as a reference, where both forms of RNAPII were present ($p<0.01$) at the promoter.

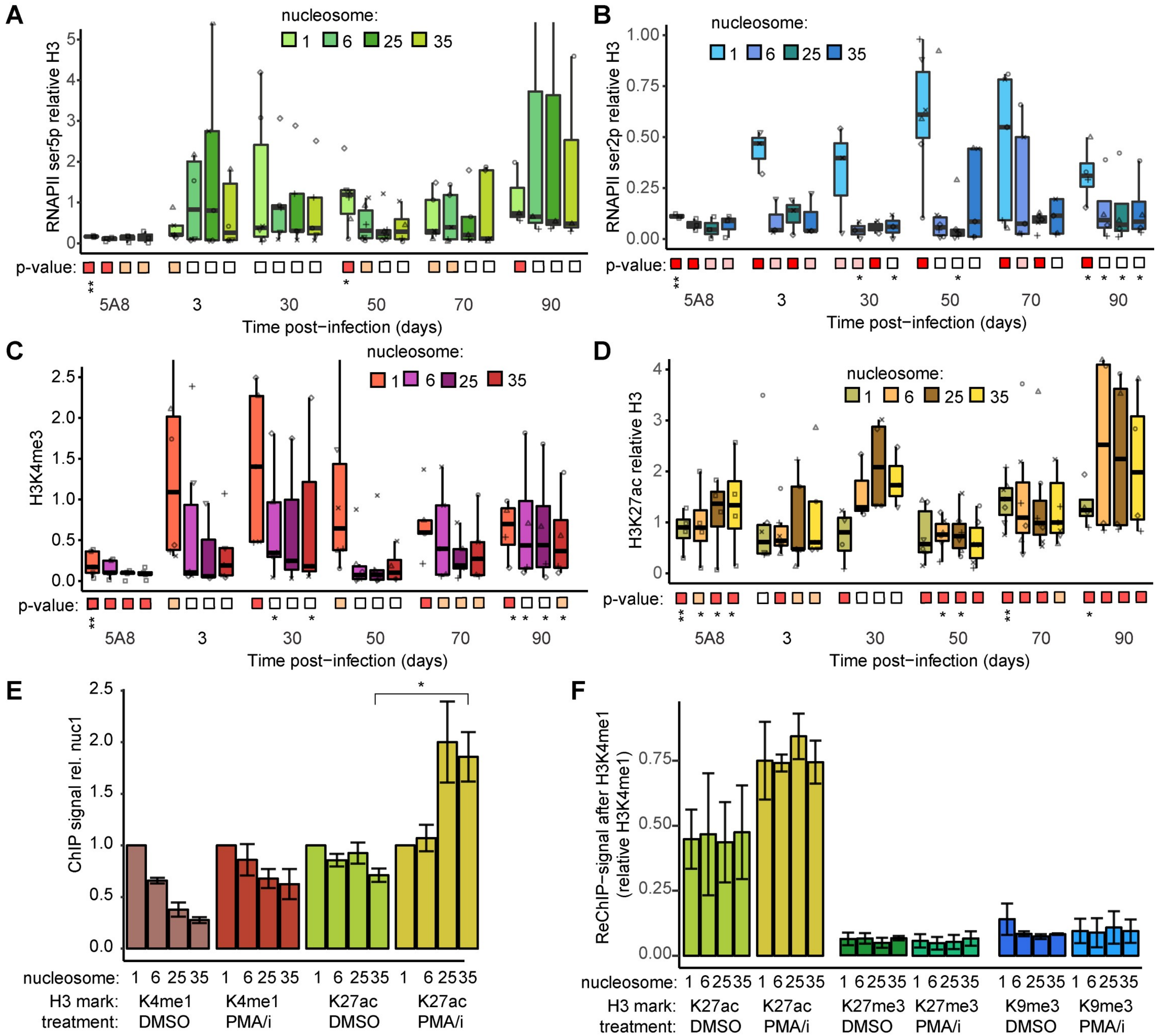


**Fig 6. RNAPII and active marks remain along HIV-1 proviral chromatin.** (**A**) ChIP-qPCR signal for initiated RNA Pol II (CTD ser5 phosphorylated). (**B**) ChIP-qPCR signal for elongating RNA Pol II (CTD ser2 phosphorylated). (**C**) ChIP-qPCR signal for promoter mark H3K4me3 **(D)** ChIP-qPCR signal for enhancer mark H3K27ac. Boxplots show data of T-cells from HIV-negative donors (*n* = 3, in duplicate). (A-D) Statistics calculated as in Fig 5A and 5B. (**E**) ChIP-qPCR for H3K4me1 and H3K27ac in 5A8 cells, following 24h DMSO or PMA/i treatment (*n* = 3), $^{*}p<0.05$, unpaired t-test. (**F**) ReChIP in 5A8 cells of H3K4me1 followed by H3K27ac, H3K27me2 or H3K9me3 at 24h post-treatment with DMSO or PMA/i (*n = 2*).


## Active chromatin marks on the provirus

We then profiled marks of active chromatin in our primary cells. The promoter mark H3K4me3 was associated with the LTR promoter during the course of the experiment as expected (Fig 6C). Strikingly, another mark, H3K27ac, which associates with active promoters and enhancers, was found throughout the provirus as the most consistent mark with the

highest ChIP signal of the tested histone marks (Fig 6D). In contrast to the H3K4me3 signal predominantly located at the promoter, the H3K27ac signal spread throughout the provirus. To test if the H3K27ac signal in fact reflected an enhancer-like chromatin [59], we also determined the levels of the enhancer mark H3K4me1. Both marks were present in resting 5A8 cells, although with different distributions, revealing an active enhancer function of the latent HIV-1 provirus (Fig 6E). Activation of the 5A8 cells had little effect on the H3K4me1 mark but led to an increase in the H3K27ac chromatin mark (Fig 6E), as previously observed in another latently HIV-infected Jurkat derived cell line [60].

To determine if the enhancer chromatin is found in a discrete subset of cells we performed sequential ChIP (re-ChIP) in 5A8 cells. Before decrosslinking, the eluate of an initial ChIP using an anti-H3K4me1 antibody, was followed by a second ChIP with antibodies against H3K27ac, H3K27me3 or H3K9me3. Through re-ChIP we could determine co-localization of histone marks at the same provirus in the same cell. As expected H3K4me1 associated strongly with H3K27ac, but not with heterochromatin marks H3K27me3 or H3K9me3 (Fig 6F). Also after T-cell activation by PMA/i, the chromatin stratification was not affected in 5A8 cells. This demonstrated that the enhancer marked provirus was present in a subpopulation of cells distinct from the heterochromatin provirus.

## Enhancer-like chromatin maintains provirus in a latent but poised state

We then asked how the proviral chromatin landscape changed during T-cell activation of primary cells. HIV-1 Bcl2-cells at 30 dpi were stimulated (αCD3-CD28) or mock-treated (DMSO). After 48 hours chromatin and DNA were isolated and we interrogated H3K27ac, H3K27me3, and H3K9me3 levels with ChIP-qPCR (Fig 7A). The stable heterochromatin marks, H3K9me3 and H3K27me3, remained unchanged in the provirus. Surprisingly, the levels of the active enhancer mark, H3K27ac, dropped significantly at the four proviral positions ($p<0.05$). This is in contrast to the observed increase in H3K27ac after T-cell activation of 5A8 cells (Fig 6E).

We subjected the same H3K27ac ChIP samples to genome-wide sequencing. Comparison to published datasets validated the baseline H3K27ac levels (S6 Fig, S2 Table). Based on ENCODE ChIP profiles of H3K4me1, H3K4me3 and H3K27ac in resting primary $CD4^{+}$ T cells, we identified 500bp regions stratified as 'enhancers' (H3K4me1+) and 'promoters' (H3K4me3+) that were either active (H3K27ac+) or poised (H3K27ac–). These regions were mapped onto our genome-wide data of H3K27ac levels before and after T-cell activation (Fig 7B). The analysis revealed that the log ratio of H3K27ac ChIP signal in activated/resting cells at both active and poised enhancers fluctuated around zero, and that the bulk of the increase in H3K27ac after T-cell activation was found in poised promoters. The loss of H3K27ac at the provirus in primary cells suggests that in these cells the HIV-1 LTR resembles an enhancer or an active promoter. However, in 5A8 cells, the HIV-1 LTR follows the expected pattern of a poised promoter.

In order to determine a role of the possible enhancer chromatin, we inhibited the histone acetyltransferase CBP/P300 by a drug GNE049 that specifically act on CBP/P300 at enhancers [61]. For technical reasons, we again turned to 5A8 cells. Even though the majority of HIV-1 LTR act as promoter, we have validated a parallel role for enhancer chromatin in these cells (Fig 6F). 5A8 cells were exposed to GNE049 3 hours prior to PMA/i addition. We used two concentrations of GNE049: 0.5μM that only inhibits CBP/P300, and 10μM that also inhibits BRD4 [62]. We confirmed that the drug decreased the level of H3K27ac in 5A8 cells (Fig 7C). HIV-1 latency reversal was scored by GFP-positive cells using flow cytometry. The GNE049 compound had a limited effect on latency reversal on its own, as only 10μM and 48 hours

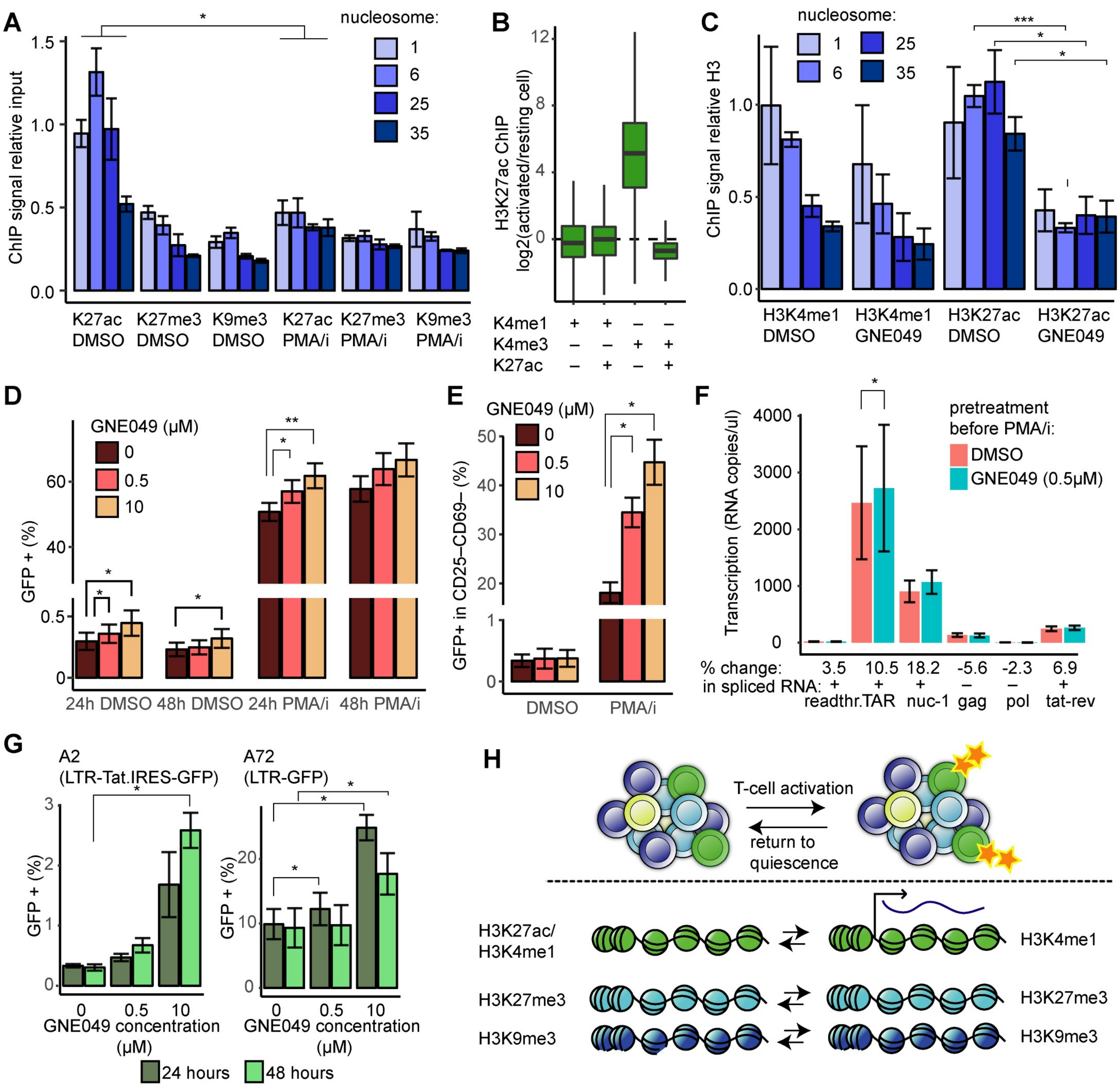


**Fig 7. Enhancer H3K27ac modulates T-cell stimulation and HIV-1 latency reversal.** (**A**) H3 modifications H3K27ac, H3K27me3 and H3K9me3 ChIP over the HIV-1 provirus. DNA and chromatin was isolated 48h after DMSO or PMA/i treatment of Bcl2-donor cells. Experiments performed in triplicate from a single donor, error bars represent standard deviation. (**B**) H3K27ac ChIP seq of the same sample as in A. Box plot of ratios for ChIP signal activated/resting cells in 500bp regions classified, based on independent published datasets, as enhancers (H3K4me1) or promoters (H3K4me1) either active (H3K27ac+) or poised (H3K27ac-). (**C**) H3K3me1 and H3K27ac ChIP signal over the provirus in 5A8 cells after treatment with GNE049 or DMSO for 3h. (**D**) HIV-1 latency reversal as measured by flow cytometry after T-cell activation in 5A8 in combination with CBP/P300 inhibitor GNE049 ($n = 6$). (**E**) Effect of GNE049 on latency reversal in cells without surface markers for activated cells ($n = 2$). (**F**) The effect of GNE049 on transcription in 5A8 cells as measured by RT-ddPCR (**G**) HIV-1 latency reversal as measured by flow cytometry after T-cell activation in A2 (LTR-Tat-IRES-GFP) and A72 (LTR-GFP) cells in combination with CBP/P300 inhibitor GNE049 (**H**) Graphical summary of the model with a heterogeneous HIV-1 reservoir of functionally intact latent proviruses and selective and transient activation of the provirus following T-cell activation. In panels C–G error bars represent s.e.m., in all panels * $p<0.05$, ** $p<0.01$, *** $p<0.001$ after unpaired t-test.


induced a modest but significant increase (39%, $p<0.05$) in GFP-positive cells. Also in combination with T-cell activation, the effect of CBP/P300 inhibition in 5A8 cells was modest but significant ($p<0.01$) (Fig 7D). Interestingly, this slight increase in number of GFP positive cells was couple with a considerably lower GFP intensity (S7A Fig). We also observed that GNE049 slightly increased the viability of the stimulated cells (S7B Fig). As T-cell activation in itself induces cell death, we stained the cells for activation surface markers CD69 and CD25. Paralleled with the viability data suggesting a lack of T-cell activation after GNE049 exposure, we observed a GNE049 dose-dependent decrease of double positive cells and an increase in double negative cells at 24 hours post PMA/i treatment ($p<0.01$) (S7C Fig). When investigating the GFP intensity in cells stratified by CD25 and CD69 surface markers, it became clear that the cellular activation status significantly ($p<0.001$) predicted the level of GFP (S7D Fig). This led us to investigate the latency reversal in the CD69-CD25- cells. No effect of GNE049 was observed in the DMSO treated cells. However, when combined with PMA/i, the GNE049 effect was striking in that the compound increased the fraction of GFP-positive cells in a dose-dependent manner from 18±2% (DMSO) to 34±3% (0.5μM GNE049) and then 45±5% (10μM GNE049) ($p<0.05$, Fig 7E).

We also determined the effect of GNE049 on transcription. Although no large changes were noted, a slight but statistically significant ($p<0.05$) increase in TAR transcription was observed (Fig 7F). Interestingly as enhancer transcription does not result in RNA splicing in contrast to mRNA transcription, only the regions present in the spliced transcript were associated with increase RNA levels, i.e. not *gag* and *pol*.

Although the GNE049 molecule specifically inhibits CBP/P300-mediated H3K27ac at enhancers in chromatin, it may also have other targets. Tat has been reported to be acetylated by CBP/P300 [63,64]. To distinguish the contribution of Tat on the latency reversal effect of GNE049, we used another HIV-1 latency model, the cells A2 and A72, where the LTR promoter is linked directly to GFP (A72), or LTR controls both GFP and Tat simultaneously [24]. In both of these cell lines, GNE049 alone increased the proportion of GFP+ cells (Fig 7G). This result demonstrates that a GNE049 effect on Tat cannot be responsible for the increased latency reversal. GNE049 had no effect on the level of GFP expression in unstimulated cells (S7E Fig). Stimulation with PMA/i induced GFP in 94% of the A2 and 85% of the A72 cells, leading to only marginal additional GFP+ cells with GNE049 (S7F Fig) (98% of A2, 88% of A72). However, we observe a GNE049-dependent increase in GFP intensity in A72 cells. This increase is not observed in Tat-containing A2 cells, suggesting that the GFP amplitude changes may be attributed to Tat. This is consistent with the know positive feed-back function of Tat.

In summary, H3K27ac at the latent provirus prevented HIV-1 activation and by removing the active enhancer functionality, more not-fully activated cells started expressing HIV-1, albeit at low levels.

## Discussion

This study demonstrated that the reservoir of latently HIV-1 infected cells comprised proviruses with heterogeneous modes of silencing, with highly different reactivation potential. From a functional cure perspective, the reservoir that requires elimination is the fraction that can be reactivated, as this is the only compartment capable of a *de novo* spread of virus. Our data suggested these reactivatable proviruses to be, at least partly, contained in active enhancer-like chromatin. This chromatin has H3K4me1 and H3K27ac modifications, and also RNAPII associated with it, but still the generated RNA fail to mature into mRNA. High levels of H3K27ac has been shown to promote general RNAPII "promoter escape" [65] and for HIV-1, H3K27ac levels correlate with proviral reactivation potential [26]. The proviral transcripts

[13,42–44] detected in quiescent T cells demonstrate the characteristics of enhancer RNA (eRNA), as they are relatively short, unspliced and non-polyadenylated. Alternatively, these unspliced transcripts might represent the full 10kb viral RNA genome, though this is unlikely as the transcribed products diminish with the distance to the TSS. eRNAs are found at active enhancers and serve to maintain open chromatin and RNAPII occupancy [66]. By using the CBP/P300 inhibitor GNE049 we could deplete H3K27ac specifically at enhancers. The resulting set of experiments showed that H3K27ac at enhancers facilitates T-cell activation HIV-1 expression. This effect is potentially mediated by indirect factors such as increasing the availability of transcription factors (Fig 7D and 7E). The observed GNE049-induced decrease of HIV-1 expression might be explained by loss of CBP/P300-mediated Tat acetylation [63,64]. At the HIV-1 locus, H3K27ac appears to prevent viral protein production. We speculate that, while the active enhancer status inhibits heterochromatin silencing of the provirus, it also prevents processive RNA transcription and mRNA splicing. At the HIV-1 provirus both H3K4me1 ("enhancer") and H3K4me3 ("promoter") are found in primary cells (Fig 6), and by reducing the enhancer activity, the balance may shift towards promoter-like transcription, which allows processive RNAPII and mRNA maturation in conjunction with other stimuli such as T-cell activation. The finding that the latent but reactivatable HIV-1 provirus act as an enhancer was supported by previous studies showing that proviruses with potential to reactivate were largely integrated into H3K4me1 and H3K27ac marked regions [22,23]. Also, HIV-1 proviruses with enhancer-like chromatin have been located in proximity of the nuclear pore, allows rapid nuclear RNA export once transcription occurs [67].

The heterochromatin marks H3K9me3 and H3K27me3 remained constant after T-cell activation. These epigenetic marks have been thoroughly described in the context of latent HIV-1, and several cofactors have been identified that initiate the epigenetic silencing [28–37,68]. The observed compaction of proviral heterochromatin (Fig 5C) is indicative of HP1α-binding [58]. Previously, in the HIV-1 promoter in the U1 cell line stimulated with PMA, it has been shown that H3K9me3 and most associated components are decreased, except for HP1α [68]. Among the members of the HP1 family, HP1α promotes condensation and phase separation [58]. Phase separation establishes and maintains distinct stable chromatin compartments. Heterochromatin loci tend to be stable during perturbations [38,69–71]. Interestingly, the resulting heterochromatin droplets are dissolved by P300-mediated histone acetylation [72]. It is tempting to speculate that the non-reactivatable HIV-1 provirus is contained within a phase-separated liquid droplet, but that H3K27ac prevents this compartmentalization to maintain the reactivation potential.

Importantly, also non-epigenetic factors contribute to the latent state of the provirus. These factors include but are not limited to transcription factors availability [16,18], transcriptional interference [15,17,19] and spatial constraints [14]. As 2–16% of the Bcl2-cells contained HIV after 50 days of culture (Fig 3B) and less than 0.2% were reactivated, even after strong stimulation by PMA/i, the vast majority of infected cells must be resistant to latency reversal. Intact proviruses being refractory to activation is an important issue as, if translated to patients, the latent reservoir as measured by replication-competent proviruses highly overestimates the number of cells that truly reserves latency.

Based on our data, we propose a simplistic model (Fig 7H) where the latent reservoir of intact proviruses is partitioned into several compartments. Separate heterochromatin compartments consist of H3K9me3 or H3K27me3 marked histones and these are governed by distinct latency reversal mechanisms. An active compartment is poised for activation characterized by enhancer marks H3K4me1 and H3K27ac, Upon T-cell stimulation, proviruses functioning as active enhancers switch from non-productive enhancer transcription to mRNA transcription, resulting in the production of viral proteins and ultimately budding

viruses. We observed a removal of the H3K27ac-mark from the activated proviruses in primary cells, contrasted by a gain of the same mark in cycling 5A8 cells. In light of Fig 7B, we speculate that this could be a consequence of the provirus to a large extent acting as a promoter in 5A8 cells but as an enhancer in primary cells. The loss of H3K27ac is consistent with other findings of RNAPII transitioning during mRNA transcription [65]. Reduction of H3K27ac during transcription activation may be a consequence of active enzymatic removal by so called "erasers", or histone replacement [71].

The low levels of HIV-1 transcripts that are prevented by H3K27ac are easily returned to latency [73]. Reestablishment of proviral latency rescues infected cells from cytopathic effects of the virus. In contrast to the well-characterized positive feedback loop mechanism driven by Tat, the mechanisms that actively silence the provirus are less studied. Recent reports have revealed negative HIV-1 feedback loops that rely on RNA precursor export [74] or histone modifications through the arginine methyltransferase CARM1 [60]. Interestingly, H3K27ac promotes CARM1-mediated HIV-1 latency. A rapid transient pulse of HIV-1 followed by programmed silencing may reseed the infection, but prevent cytopathic effects of the virus and prevent immunological detection.

Our findings pinpointed some discrepancies among model systems of HIV-1 latency. Although the Bcl2-system allows the study of prolonged HIV-1 latency in resting cells, it suffers from the reduced cell viability of the cultures. Also, cells were analyzed after maximal stimulation. *In vivo*, as in our experiment, this strongly affects the viability. The fact that cell death correlates with HIV-1 activation [75] adds a confounding variable, especially to the LRA experiments. Interestingly though, the LRA experiments revealed discrepancies in that the Bcl2-cells with established HIV-1 latency showed little proviral activation from the HDAC inhibitors. The same HDAC inhibitors were the most promising LRAs in newly infected $T_{EM}$ and or cycling 5A8 cells. Broad HDAC inhibitors result in an overall increase in histone acetylation, including the H3K27ac mark. This may explain why H3K27ac has opposing roles in over-all HIV-1 activation in 5A8 cells and latent Bcl2-cells. Consequently, it may also partly explain the limited effect of HDAC inhibitors *in vivo*.

Further, the primary cultures with Bcl2-cells only contain a minority of cells encoding the *bcl2* construct. The *bcl2* construct uses the same 5´ LTR promoter sequence as the HIV-1 construct and throughout the time course we can only detect the 5´LTR in less than 7% of the cells. This means that the majority of the cells are not modified with the prosurvival gene, but instead are protected from cell death by other means. Possibly the minority of *bcl2*+ cells act as isogenic feeder-cells that sustain the cultures.

Previous models have shown that a medication-free sustained remission ("cure") of HIV-1 infection would require a reduction of the reservoir by >4 log units [76]. However, the heterogeneous reservoir mainly consists of non-functional proviruses, as a result of mutations and epigenetic silencing. Instead of aiming to eradicate the entire proviral pool, a functional cure only requires the removal of the fraction of the reservoir with reactivation potential. The data presented here suggest that this fraction might be constituted of proviruses with an enhancer-like chromatin structure.

## Materials and methods

### Ethics statement

This study was approved by the Ethics Committee (Regionala Etikprövningsnämnden Stockholm, Reg#2017/1138-31 and Reg#2018/102-31), and written informed consent was obtained from all subjects. The data were analyzed anonymously.

## Human samples

Buffy coats from 450-ml blood samples drawn from HIV-negative donors were provided by the Karolinska Universitetslaboratoriet. The samples were anonymized before arrival. Blood samples from HIV positive participants were obtained from the HIV unit at Department of Infectious Diseases, Karolinska University Hospital.

## Cell culture

Bcl2 model cells were generated as previously described [53]. Peripheral blood mononuclear cells (PBMCs) from HIV negative and positive study participants were purified on Ficoll-paque PLUS (GE Healthcare, Cat#17-1440-02). $CD4^+$ T lymphocytes were extracted (Miltenyi Biotec Cat#130-096-533) by negative selection. Resting $CD4^+$ T cells were isolated by subsequent negative selection using CD25 MicroBeads II, CD69 MicroBead Kit II, and Anti-HLA-DR MicroBeads (Miltenyi Biotec Cat#130-092-983, Cat#130-092-355, Cat#130-046-101). Cells were kept in RPMI 1640 medium (Hyclone, Cat# SH30096_01), 10% FBS (Life Technologies, Cat# 10270–106), 1% Glutamax (Life Technologies, Cat# 35050), 1% Penicillin-streptomycin (Life Technologies, Cat# 15140–122). For active growth conditions, media was supplemented with human interleukin-2 IS (Miltenyi Biotec, Cat#130-097-742; Lot#5170516373) final concentration 100U/ml and 5% T-cell conditioned media, according to the protocol.

## Virus production

EB-FLV (containing *bcl2*), pNL4-3-Δ6-drEGFP (reporter HIV-1), pHelper [54], and pMD2.G (VSV-g) (Addgene, Cat#12259) plasmids were purified with Plasmid Plus Maxi Kit (Qiagen, Cat# 12963). 293T cells (ATCC, CRL-3216; CVCL_0063) grown in DMEM media (Hyclone, Cat# SH30022_01) were transfected with Lipofectamine LTX with PLUS reagent (Thermo-Fisher, Cat# 15338100), and, after an additional 48 h, supernatants were harvested. We tested the functional infectivity of NL4-3-Δ6-drEGFP by transducing 293T cells (American Type Culture Collection, Cat# CRL-3216) with the viral particles. After 48 h, we measured GFP signals with flow cytometry. We determined virus titers by the HIV-1 p24 ELISA Assay (XpressBio, Cat# Cat#XB-1000). Virus-containing supernatant was concentrated with LentiXconcentrator (Clontech, Cat# 631231).

## Lentiviral infection

Prior to infection, cells were activated for 72 h in media with 1 μg/ml anti-CD28 (BD, Cat# 555725) in 6-well plates coated for 1 h at 37˚C with 10 μg/ml anti-CD3 (BD, Cat# 555336). Cells where then spinoculated (2h at 1,200*g* 25˚C) with pseudotyped EB-FLV or NL4-3-Δ6-drEGFP at a concentration of 250 ng p24 per $1\times10^6$ cells.

## Chemicals to induce proviral activation

Cells were exposed to latency-reversal agents for 48 h (or as indicated). Drugs and chemicals used were phorbol 12-myristate 13-acetate (Sigma-Aldrich, Cat# 79346) final concentration 50 ng/ml, ionomycin (Sigma-Aldrich, Cat# I0634; Lot#106M4015V) final concentration 1 μM, panobinostat (Cayman Chemicals, Cat# CAYM13280) final concentration 30 nM or 150nM, JQ1 (Cayman Chemicals, Cat#CAYM11187) final concentration 100nM, bryostatin (Biovision, Cat# BIOV2513) final concentration 10nM, GNE049 (MedChemExpress, Cat# HY-108435). For all treatments, raltegravir (Sigma-Aldrich, Cat# CDS023737) was added to the medium at final concentration 2 μM.

## Flow cytometry

Cells were stained with mouse anti-human CD25 APC (clone M-A251, BD 560987); CD69 PE-Cy7 (clone FN50, 557745), LIVE/DEAD Fixable Violet Dead Cell Stain (ThermoFisher, Cat# L34955), and fixed in 2% formaldehyde for 30 min. Flow analysis was performed on a CytoFLEX S (Beckman Coulter). Individual flow droplets were gated for lymphocytes, viability, and singlets. Data was analyzed by Flowjo 10.1 (Tree Star).

## Isolation of nucleic acids

Genomic DNA and RNA from PLWH cells were isolated simultaneous by Allprep DNA/RNA MiniKit (Qiagen, Cat#80204). In cases of low yield, RNA samples were concentrated using RNA Clean and Concentrator (Zymo Research, Cat#R1013). Genomic DNA from Bcl2-cells and 5A8 were isolated from the input samples of the chromatin preparations. Total cellular RNA from Bcl2-cells and 5A8 was isolated from 1–5×$10^6$ cells with RNeasy Mini Plus Kit (Qiagen, Cat#74134).

## Measurement of intracellular HIV-1 transcripts

RNA (100–500ng) was used directly in reactions with SuperScript III Reverse Transcriptase (Invitrogen, Cat#11752–050), primed by random hexamers (ThermoFisher, Cat#S0142). Reactions were incubated at 25˚C for 10 min, followed by 50˚C for 30 min. Reactions were terminated at 85˚C for 5 min followed by incubation on ice. Subsequently, 2U/reaction of *E.coli* RNAse H (Invitrogen, Cat#18021–014) was added and tubes were left at 37˚C for 20 min, after which they were stored at -20˚C. cDNA was specifically quantitated at specific positions with ddPCR (S3 Table).

## Droplet digital PCR (ddPCR)

ddPCR was performed with the QX200 Droplet Digital qPCR System (Bio-Rad). Samples were tested in duplicate, and each reaction consisted of a 20-μl solution containing 10 μl Supermix for Probes without dUTP (Bio-Rad, Berkeley, CA, USA), 900 nM primers, 250 nM probe (labeled with HEX or FAM), and 5 μl undiluted RT product or 100–500 ng cellular DNA (fragmented with a QIAshredder column). Emulsified PCR reactions were performed with a C1000 Touch thermal cycler (Bio-Rad), with the following protocol: 95˚C for 10 min, followed by 40 cycles of 94˚C for 30 s and 60˚C for 60 s, and a final droplet cure step of 10 min at 98˚C. Each well was then read with a QX200 Droplet Reader (Bio-Rad). Droplets were analyzed with QuantaSoft, version 1.5 (Bio-Rad), software in the absolute quantification mode. When replicates were used, the percentage of mutant fractional abundance was extracted as merged samples. For visualization, we used the "twoddpcr" Bioconductor/R package [77]. "Rain" droplets were classified by fitting the clusters to bivariate normal distributions using the mahalanobisRain function followed by visual inspection and, if needed, correction. Nucleotide numbers are set according to the coordinates of the reference Human immunodeficiency virus type 1 (HXB2; K03555)

## Chromatin immunoprecipitation-PCR

Prior to chromatin extraction, viable cells were isolated using Ficoll density separation (300*g* for 10 min at room temperature). ChIP-qPCR was performed using the iDeal ChIP-qPCR Kit (Diagenode, Cat# C01010180). Each ChIP reaction was performed on 1–2 × $10^6$ cells. Sonication was performed at 30s in eight cycles (Bioruptor Pico, Diagenode, Cat# B01060010). Input samples were removed. ChIP antibodies were targeting H3 (Abcam, Cat# ab1791), H3K4me1

(Abcam, Cat# ab8895), H3K4me3 (Diagenode, Cat# C15410030), H3K9me3 (Abcam, Cat# ab8898), H3K27me3 (Diagenode, Cat#C15410069), H3K27ac (Abcam, Cat# ab4729), RNA-PII-ser2ph (Diagenonde, Cat# C15200005), RNAPII-ser5ph (Diagenode, Cat# C15200007), IgG (Diagenode, Cat# C15410206). ChIP eluates were purified with Wizard SV Gel and PCR clean-up system (Promega, Cat# A9282). Primer sequences are shown in S3 Table. PCR reactions were performed with Powerup Sybr green master mix (2x) (ThermoFisher, Cat#A25742) using 40 cycles on an Applied Biosystems 7500 Fast Real-Time PCR System (ThermoFisher).

For each datapoint the specific ChIP signal was divided by the H3 signal for the same sample. Samples where the unspecific IgG signal was higher than the specific signal were discarded. P-values were calculated both using a one-sample t-test (null hypothesis: specific normalized ChIP signal equals zero), and using a two-sample t-test (null hypothesis: specific ChIP signal equals IgG signal). Significance threshold were set at $p<0.05$ and $p<0.01$. For the boxplot visualizations, the median signal is represented by the thick line and the box covers the 25 to 75 percentile of the values. The individual data points were visualized by differently shaped points.

## Sequential chromatin immunoprecipitation (Re-ChIP)

Re-ChIP was performed using a modified iDeal ChIP-qPCR Kit (Diagenode, Cat# C01010180). In short, each ChIP reaction was performed on $1 \times 10^7$ cells. Sonication was performed at 30s in ten cycles (Bioruptor Pico, Diagenode, Cat# B01060010). ChIP antibodies were targeting H3K4me1 or IgG (same as above). First bead elution was done in 10 mM Tris–HCl (pH 8.0), 2 mM EDTA,10 mM DTT, for 30 min at 37˚C. The eluate was used for a second ChIP using antibodies targeting H3K9me3, H3K27me3, H3K27ac, and IgG. ChIP eluates here were purified with standard buffers and used for qPCR.

## Chromatin accessibility

Nuclease accessibility was evaluated through the Chromatin Accessibility Assay Kit (Abcam, Cat# ab185901) according to manufacturer's instructions. Per reaction, $0.5 \times 10^6$ cells were used.

## Sequencing

DNA samples were quantified with Qubit dsDNA HS Assay kit (ThermoFisher, Cat# Q32851) and libraries were prepared using NEBNext Ultra II DNA library kit they were sequenced on an Illumina Hiseq 2000 (50 cycles, single-end sequencing, 50 bases) at the BEA facility (Huddinge, Sweden), according to the manufacturer's instructions. Raw data from the Hiseq (fastq files) were aligned to the hg19 genome assembly with the Bowtie2 program (version 2.0.6), set to the default parameters. Resulting sam files were converted to bam files using Samtools version 1.4. Bam files were imported into SeqMonk version 0.33.0 where 2kb probes were constructed around the 5′ position of all 40,147 genes of the GRCh37 assembly. Probes were quantitated with 'Read Count Quantitation' using 'All Reads' correcting for total count per million reads, duplicates were ignored.

RNA-seq (mRNA) data from primary $CD4^+$ cells were collected from GSM669617 (GEO). ENCODE ChIP-seq data for comparison were collected from ENCFF618IUD and ENCFF862SKP (H3K27ac), ENCFF499NFE, ENCFF989BNS, ENCFF112QDR (H3K4me1), and ENCFF158NCC, ENCFF942WXX (H3K4me3). Regions of 500bp were designed around the Eponine complete set of 61,153 transcription start sites (TSS). Number of reads per million total reads for each region was calculated. The TSS regions were designated to groups based on above median signal for the three marks. Active enhancer: above median for H3K27ac and

H3K4me1, low H3K4me3 (n = 105); active promoter: above median for H3K27ac and H3K4me3, low H3K4me1 (n = 22,506); Active enhancer: above median for H3K27ac and H3K4me1, low H3K4me3 (n = 105); active promoter: above median for H3K27ac and H3K4me3, low H3K4me1 (n = 22,506);. These regions were then mapped onto ChIP-seq from patient 2 and the reads were quantitated. Number of reads per million total reads and the ratio ($H3K27ac_{activatied}/input_{activated}$)/ ($H3K27ac_{resting}/input_{resting}$) were calculated for each region.

## Supporting information

**S1 Fig. HIV-1 primers specifically detected both CA-RNA and genomic DNA in cells from HIV-1 positive study participants.** (**A**) Primer positions relative to the HIV-1 provirus. Viral proteins depicted with blue bars. (**B**) RT-ddPCR using primers for HIV-1 tested on RNA from CD4 cells isolated from HIV-1 negative (HIV−) (*n* = 4) and HIV-1 positive (HIV+) (in total *n* = 10, here represented by *n* = 2) study participants, values normalized to 500ng RNA. Axis is broken to depict that for all but the *pol* and *nef* primer probe combinations, no HIV-1 signal was consistently detected in cells from HIV-1 negative donors (**C**) ddPCR of DNA from the cells originating from HIV-1 positive study participants. Probe efficiency identified cells where the primer-probe combinations were able to detect genomic proviral DNA (n = 10). (**D**) RT-ddPCR on RNA isolated from CD4-depleted T-cells from HIV-1 positive study participants (*n* = 5).
(TIF)

**S2 Fig. PMA/i increases the intensity of GFP among a panel of LRAs.** Cells were exposed to different agents for 24 hours, after which cells were fixed and the intensify of GFP among the GFP-positive cells were recorded by flow cytometry. 5-azadC in all conditions (alone or in combination) was added 72 hours prior to fixation. (**A**) *In vitro* HIV-1 infected $T_{EM}$ cells from HIV-1 negative study participants (*n* = 4) and (**B**) J-lat 5A8 cells (*n* = 2, technical replicates).
(TIF)

**S3 Fig. Cell viability and proviral expression in primary *in vitro* HIV-1 infected Bcl2 cells.** (**A**) Viability of Bcl2-cultures as measured by a membrane-permeable dye (*n* = 3). (**B**) CA-RNA levels originating from the provirus were quantified by RT-ddPCR in primary HIV-1 Bcl2 cells at 50 dpi. Probes were as in previous results (S1A Fig) (*n* = 2).
(TIF)

**S4 Fig. T-cell activation leads to a modest upregulation of cell surface markers CD25 and CD69 in primary cells.** J-lat 5A8 cells (upper panels) and example of primary Bcl2 cells with HIV-1-GFP at 50 dpi (lower panels) were exposed to DMSO (left), antibodies against CD3 and CD28 (middle), or PMA/ionomycin (right) for 48 hours prior to flow cytometry analysis using labeled antibodies against surface markers CD25 and CD69.
(TIF)

**S5 Fig. Cell viability after drug exposure.** Boxplot showing the cell viability as determined by membrane integrity through LIVE/DEAD staining and flow cytometry. HIV-1 infected Bcl2 model cells from healthy donors (*n* = 3) were exposed to drugs for 48h and 72h. J-lat clone 5A8 was used as control.
(TIF)

**S6 Fig. Comparison between H3K27ac ChIP in HIV-1-GFP infected Bcl2-model cells from healthy donor and ENCODE dataset.** Boxplot showing the H3K27ac ChIP signals (resting CD4+ T-cells) calculated in 2kb-probes centered around the start of genes. Published ChIP data (ENCODE ENCFF862SKP) were processed in the same way and grouped in quartiles. All

individual data points are shown.
(TIF)

**S7 Fig. The effect of GNE049 on viability and GFP intensity.** (**A**) GFP intensity in 5A8 GFP + cells treated with GNE049. (**B**) Cell viability of activated 5A8 cells increase after treatment with GNE049. Cells were exposed to GNE049 or DMSO for 3 hours prior to treatment with PMA and ionomycin (PMA/i) or DMSO. After 24 or 48 hours, cells were stained with a LIVE/DEAD membrane-permeable dye and fixed; thereafter cells were analyzed by flow cytometry ($n$ = 7). (**C**) The appearance of surface markers for activated cells (CD25 and CD69) after T-cell activation and GNE049 treatment ($n$ = 2). (**D**) GFP intensity of cells in panel C. (**E**) In A2 and A72 cells, GFP intensity after GNE049 treatment (3h) followed by DMSO for 24 or 48h (**F**) In A2 and A72 cells, percentage of GFP positive cells and GFP intensity after GNE049 treatment (3h) and stimulation by PMA/i for 24 or 48h. $^{*}p<0.05$, $^{**}p<0.01$ paired t-test.
(TIF)

**S1 Table. Characteristics of HIV-positive study participants.** Abbreviations: 3TC, lamivudine; ABC, abacavir; CAB, cabotegravir; CFR, circle recombinant form; Cobi, cobicistat; DRV, darunavir; DTG, dolutegravir; EFV, efavirenz; EVG, elvitegravir; F, female; FTC, emtricitabine; M, male; ND, Not determined; RPV, rilpivirine; r, ritonavir-boosted; TAF, tenofovir alafenamide; TDF, tenofovir disoproxil fumarate.
[1] Patient 1 started treatment 5 months 2014 but it was interrupted and started again in Sep 2016. Patient 4 started treatment 1996–1999 but was interrupted between 1999–2000. Patient 8 had a treatment interruption.
[2] One blip in Viral Load in 2014
[3] 8 days before sampling.
(XLSX)

**S2 Table. H3K27ac ChIP and input signal over 2kb probes of genes.** Quantification of sequencing reads overlapping 40,147 probes designed around the 5′ region of genes in the GRCh37 assembly.
(XLSX)

**S3 Table. Oligonucleotide sequences and positions relative to the HXB2 reference genome.**
[*] not corresponding to the exact sequence of the HXB2 reference genome
RC: reverse complement.
(XLSX)

## Acknowledgments

The following reagent was obtained through the NIH AIDS Reagent Program, Division of AIDS, NIAID, NIH: J-Lat Tat-GFP Cells and J-Lat GFP Cells from Dr. Eric Verdin. J-Lat 5A8 cells were kindly provided by Dr. Eric Verdin. The plasmids, pCM6 and pC-Help, were gifts from Robert Silicano, and the pMD2.G plasmid was a gift from Didier Trono (Addgene plasmid # 12259). We thank Andreas Lennartsson for critical reading of the manuscript. We would like to acknowledge the core facilities MedH Core Flow Cytometry facility (Karolinska Institutet) for providing cell analysis services, and BEA, Bioinformatics and Expression Analysis (Karolinska Institutet) for providing sequencing services.

## Author Contributions

**Conceptualization:** J. Peter Svensson.

**Funding acquisition:** Anders Sönnerborg, J. Peter Svensson.

**Investigation:** Birgitta Lindqvist, J. Peter Svensson.

**Resources:** Sara Svensson Akusjärvi, Anders Sönnerborg, Marios Dimitriou.

**Supervision:** J. Peter Svensson.

**Writing – original draft:** J. Peter Svensson.

**Writing – review & editing:** Birgitta Lindqvist, Sara Svensson Akusjärvi, Anders Sönnerborg, Marios Dimitriou, J. Peter Svensson.

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
