## [Decision Letter · Decision Letter 0]

27 Nov 2019

Dear Dr. Svensson:

Thank you very much for submitting your manuscript "Chromatin maturation of the HIV-1 provirus in primary resting CD4+ T cells" (PPATHOGENS-D-19-01861) for review by PLOS Pathogens. Your manuscript was fully evaluated at the editorial level and by independent peer reviewers. The reviewers appreciated the attention to an important topic but identified some aspects of the manuscript that should be improved.

We therefore ask you to modify the manuscript according to the review recommendations before we can consider your manuscript for acceptance. Your revisions should address the specific points made by each reviewer.

(1) A letter containing a detailed list of your responses to the review comments and a description of the changes you have made in the manuscript. Please note while forming your response, if your article is accepted, you may have the opportunity to make the peer review history publicly available. The record will include editor decision letters (with reviews) and your responses to reviewer comments. If eligible, we will contact you to opt in or out.

(2) Two versions of the manuscript: one with either highlights or tracked changes denoting where the text has been changed; the other a clean version (uploaded as the manuscript file).

We hope to receive your revised manuscript within 60 days or less. If you anticipate any delay in its return, we ask that you let us know the expected resubmission date by replying to this email.

[LINK]

Sincerely,

David T. Evans

Associate Editor

PLOS Pathogens

Thomas Hope

Section Editor

PLOS Pathogens

Kasturi Haldar

Editor-in-Chief

PLOS Pathogens

orcid.org/0000-0001-5065-158X

Grant McFadden

Editor-in-Chief

PLOS Pathogens

orcid.org/0000-0002-2556-3526

Both reviewers felt that the revised manuscript addressed most of their previous concerns and is substantially improved, but raised a few remaining questions. Please address these remaining points as best you can given the limitations of your model system.

Reviewer's Responses to Questions

**Part I - Summary**

Reviewer #1: PLOS Pathogens

Chromatin maturation of the HIV-1 provirus in primary resting CD4+ T cells (Birgitta Lindqvist, Sara Svensson Akusjärvi, Anders Sönnerborg, Marios Dimitiou, J. Peter Svensson) (PPATHOGENS-D-19-01861)

The authors added comments, corrected or performed additional experiments to address the majority of the reviewers concerns. However, some points are still not convincing and need to be further addressed.

Reviewer #2: Overall, it seems to me that the authors have addressed most of the questions raised by reviewers, though some caveats in this study remain and could be discussed.

The major caveats of the primary cell latency model include a low basal level of GFP+ cells and low induction suggested by GFP+ cell rate, which was pointed out by reviewers in the major suggestions. These caveats are in the nature of the bcl-2 primary cell model. They were well addressed by the authors by revising the text and explained the data more extensively. Increasing the number of donors and substantial statistical analysis might also have helped to clarify the theory. However, it’s noteworthy that the caveats associated with the model do significantly weaken the conclusions. The initial frequency of infected cells was around 10% (line 218.) After weeks of culturing, GFP+ rates decreased to 1%, which was lower than 2% even upon stimulation. It suggests that 80-90% of HIV provirus were resistant to the reactivation, which should be kept in mind when interpreting the data in figures 5 and 6.

I doubt that switching between Jlat and bcl2 model (especially in figure 5-7) added any value to the study other than technical convenience. Epigenetic character is heavily system dependent. I don’t think that the data using primary cells and cell lines are comparable or interchangeable.

Also, I understand epigenetic is the main focus of this study. But, other cellular factors likely contribute to the transcriptional status of provirus together with epigenetic factors, which should at least be discussed.

**Part II – Major Issues: Key Experiments Required for Acceptance**

Reviewer #1: Major comments:

1) In this work, the key conclusion that the proviruses actively transcribed following T-cell activation bear enhancer chromatin marks is based on the loss of this epigenetic mark and on a comparison with ChIP seq data from cellular genes. The reviewer 1 asked for additional experiments and controls to improve this conclusion. To address to this point, the authors used the GNE049 drug which depletes H3K27ac specifically at enhancers in 5A8 cells which have been described in Figure 8 to contain low level of H3K27ac before activation. The 5A8 cells are not the best model to conclude on the need to lose enhancer marks to reactivate HIV from latency since 5A8 cells contain high level of H3K27Ac in the provirus after PMA activation. It is therefore not surprising that increasing doses of GNE049 decreased the percentage of GFP intensity (Fig. 7D). To improve their experiments using the GNE049 drug, the authors should use a better HIV-1 latency model (in this case HIV-1 latently-infected cells with higher amount of H3K27ac enhancer mark) coupled to viral transcripts measurement.

2) No explanation is given by the authors regarding the levels of negative epigenetic marks that remained constant following TCR activation (Figure 7A). The authors responded by stating : “that heterochromatin marks at the provirus are not affected by T-cell activation as an indication that latency of these proviruses are not reversed, at least not to a level that we can reliably measure it in this experiment”. The authors should further discussed (in regard to previous work in the literature) the epigenetic marks H3K9me3 and H3K27me3 which remained constant after TCR activation and provide the percentage of GFP positive cells in absence and presence of TCR activation, supporting the absence of HIV-1 latency reversal.

Reviewer #2: no

**Part III – Minor Issues: Editorial and Data Presentation Modifications**

Reviewer #1: Minor comments:

1) Reviewer 1 mentioned that in Fig 2F: at 3dpi, a detection of all HIV-1 transcripts should be observed. However, no detection of these transcripts was observed. In Figure 2F, all HIV-1 transcripts were not observed with no detection of transcripts downstream nuc1. The authors responded to this point by stating :” This discrepancy might be due to technical issues or that transcription at 3 dpi is strongly reduced and the observed protein levels are lingering from earlier translation”. They also included a section in the Discussion on the cellular viability issue. The authors should provide information and/or additional figure regarding the cell viability. Same remark for Figure 3C at 70 days post-infection

2) Reviewer 3 observed variations throughout the proviral genome without statistical analysis of the significance of these variations over the time post-infection (Figure 5A). The authors answered by performing statistical analysis on the dataset. However, they observed statistical significance of heterochromatin accumulation at several points. Thus, the authors should be careful when stating (line 314): “The constitutive H3K9me3-mark appeared to be uniformly distributed over the proviral body by 30 dpi” and should further discuss these variations.

3) The authors should explain and discuss the contradictory results regarding the H3K27ac mark obtained in 5A8 and HIV-1 BCL2 cells (Figures 6E and 7A).

Reviewer #2: please see above

PLOS authors have the option to publish the peer review history of their article (what does this mean?). If published, this will include your full peer review and any attached files.

Reviewer #1: No

Reviewer #2: No

---

## [Editor Report · Decision Letter 1]

9 Dec 2019

Dear Dr. Svensson,

We are pleased to inform that your manuscript, "Chromatin maturation of the HIV-1 provirus in primary resting CD4+ T cells", has been editorially accepted for publication at PLOS Pathogens. 

Before your manuscript can be formally accepted and sent to production, you will need to complete our formatting changes, which you will receive by email within a week. Please note that your manuscript will not be scheduled for publication until you have made the required changes.

IMPORTANT NOTES

(1) Please note, once your paper is accepted, an uncorrected proof of your manuscript will be published online ahead of the final version, unless you’ve already opted out via the online submission form. If, for any reason, you do not want an earlier version of your manuscript published online or are unsure if you have already indicated as such, please let the journal staff know immediately at plospathogens@plos.org.

(2) Copyediting and Proofreading: The corresponding author will receive a typeset proof for review, to ensure errors have not been introduced during production. Please review the PDF proof of your manuscript carefully, as this is the last chance to correct any errors. Please note that major changes, or those which affect the scientific understanding of the work, will likely cause delays to the publication date of your manuscript. 

(3) Appropriate Figure Files: Please remove all name and figure # text from your figure files. Please also take this time to check that your figures are of high resolution, which will improve the readbility of your figures and help expedite your manuscript's publication. Please note that figures must have been originally created at 300dpi or higher. Do not manually increase the resolution of your files. For instructions on how to properly obtain high quality images, please review our Figure Guidelines, with examples at: http://journals.plos.org/plospathogens/s/figures.

(4) Striking Image: Please upload a striking still image to accompany your article if one is available (you can include a new image or an existing one from within your manuscript). Should your paper be accepted, this image will be considered for our monthly issue image and may also appear on our website to feature your article. Please upload this as a separate file, selecting "striking image" as the file type upon upload. Please also include a separate "Other" file with a caption, including credits and any potential copyright information. Please do not include the caption in the main article file. If your image is from someone other than yourself, please ensure that the artist has read and agreed to the terms and conditions of the Creative Commons Attribution License at http://journals.plos.org/plospathogens/s/content-license. Please note that PLOS cannot publish copyrighted images.

(5) Press Release or Related Media: If your institution or institutions have a press office, please notify them about your upcoming paper at this point, to enable them to help maximize its impact. If they will be preparing press materials for this manuscript, please inform our press team in advance at plospathogens@plos.org as soon as possible. We ask that you contact us within one week to plan ahead of our fast Production schedule. If you need to know your paper's publication date for related media purposes, you must coordinate with our press team, and your manuscript will remain under a strict press embargo until the publication date and time. This means an early version of your manuscript will not be published ahead of your final version. 

(6)  PLOS requires an ORCID iD for all corresponding authors on papers submitted after December 6th, 2016. Please ensure that you have an ORCID iD and that it is validated in Editorial Manager.  To do this, go to ‘Update my Information’ (in the upper left-hand corner of the main menu), and click on the Fetch/Validate link next to the ORCID field.  This will take you to the ORCID site and allow you to create a new iD or authenticate a pre-existing iD in Editorial Manager

(7) Update your Profile Information: Now that your manuscript has been provisionally accepted, please log into Editorial Manager and update your profile, if needed. Go to https://www.editorialmanager.com/ppathogens, log in, and click on the "Update My Information" link at the top of the page. Please update your user information to ensure an efficient production and billing process. 

(8) LaTeX users only: Our staff will ask you to upload a TEX file in addition to the PDF before the paper can be sent to typesetting, so please carefully review our Latex Guidelines http://journals.plos.org/plospathogens/s/latex in the meantime.

(9) If you have associated protocols in protocols.io, please ensure that you make them public before publication to guarantee immediate access to the methodological details.

Best regards,

David T. Evans

Associate Editor

PLOS Pathogens

Thomas Hope

Section Editor

PLOS Pathogens

Kasturi Haldar

Editor-in-Chief

PLOS Pathogens

orcid.org/0000-0001-5065-158X

Grant McFadden

Editor-in-Chief

PLOS Pathogens

orcid.org/0000-0002-2556-3526
---

## [Editor Report · Acceptance letter]

30 Dec 2019

Dear Dr. Svensson,

We are delighted to inform you that your manuscript, "Chromatin maturation of the HIV-1 provirus in primary resting CD4+ T cells," has been formally accepted for publication in PLOS Pathogens.

Best regards,

Kasturi Haldar

Editor-in-Chief

PLOS Pathogens

orcid.org/0000-0001-5065-158X

Grant McFadden

Editor-in-Chief

PLOS Pathogens

orcid.org/0000-0002-2556-3526